# Cost-Effective Protein Production in CHO Cells Following Polyethylenimine-Mediated Gene Delivery Showcased by the Production and Crystallization of Antibody Fabs

**DOI:** 10.3390/antib12030051

**Published:** 2023-08-04

**Authors:** Klaudia Meskova, Katarina Martonova, Patricia Hrasnova, Kristina Sinska, Michaela Skrabanova, Lubica Fialova, Stefana Njemoga, Ondrej Cehlar, Olga Parmar, Petr Kolenko, Vladimir Pevala, Rostislav Skrabana

**Affiliations:** 1Institute of Neuroimmunology, Slovak Academy of Sciences, 845 10 Bratislava, Slovakia; 2Faculty of Natural Sciences, Comenius University, 842 15 Bratislava, Slovakia; 3AXON Neuroscience R&D Services SE, 811 02 Bratislava, Slovakia; 4Faculty of Nuclear Sciences and Physical Engineering, Czech Technical University in Prague, 115 19 Prague, Czech Republic; 5Institute of Molecular Biology, Slovak Academy of Sciences, 845 51 Bratislava, Slovakia

**Keywords:** CHO cell line, recombinant protein production, polyethylenimine, gene delivery, production costs, protein crystallography

## Abstract

Laboratory production of recombinant mammalian proteins, particularly antibodies, requires an expression pipeline assuring sufficient yield and correct folding with appropriate posttranslational modifications. Transient gene expression (TGE) in the suspension-adapted Chinese Hamster Ovary (CHO) cell lines has become the method of choice for this task. The antibodies can be secreted into the media, which facilitates subsequent purification, and can be glycosylated. However, in general, protein production in CHO cells is expensive and may provide variable outcomes, namely in laboratories without previous experience. While achievable yields may be influenced by the nucleotide sequence, there are other aspects of the process which offer space for optimization, like gene delivery method, cultivation process or expression plasmid design. Polyethylenimine (PEI)-mediated gene delivery is frequently employed as a low-cost alternative to liposome-based methods. In this work, we are proposing a TGE platform for universal medium-scale production of antibodies and other proteins in CHO cells, with a novel expression vector allowing fast and flexible cloning of new genes and secretion of translated proteins. The production cost has been further reduced using recyclable labware. Nine days after transfection, we routinely obtain milligrams of antibody Fabs or human lactoferrin in a 25 mL culture volume. Potential of the platform is established based on the production and crystallization of antibody Fabs and their complexes.

## 1. Introduction

Production of recombinant proteins is fundamental for academic research as well as to produce pharmaceuticals in the pharma industry. Recombinant mammalian proteins including monoclonal antibodies and their fragments are the fastest growing class of biologics to treat oncological, autoimmune, and cardiovascular or neurodegenerative diseases [1,2,3,4,5]. While industries require a scalable and reproducible expression [6,7], academic research projects, on the other hand, benefit from cost-effective, fast and flexible protein production pipelines.

Correct folding and nearly native level of phosphorylation, glycosylation and other modifications can be achieved after protein expression in mammalian cell lines, namely in Human Embryonic Kidney (HEK) 293 and Chinese Hamster Ovary (CHO) cell lines [8]. Other expression systems frequently used in research are not universally applicable for mammalian proteins due to an insufficient oxidative folding capacity (bacteria) or inappropriate posttranslational modifications (yeast, plant or insect systems) [9,10].

After recent progress in cultivation and transient gene expression (TGE), CHO cells outperform the HEK293 cell line in terms of achievable yields and efficiency of gene delivery [7]. CHO line is being used as the method of choice for the development of stable expressing cell lines or stable pool generation for protein production in pharma. The yield of TGE can range from one to hundreds of milligrams per liter [6,7]. The gene delivery to the cells is performed via electroporation or through cellular uptake of encapsulated DNA. Electroporation of genes could reach more than 1 g/L production within two weeks [11]. However, this method needs special instrumentation and costly consumables and is therefore unsuitable for small research projects. Encapsulation of DNA requires the creation of positively charged lipoplexes or polyplexes. In addition to the polycationic lipid transfection reagents, the cationic polymer polyethylenimine (PEI) is the most cost-effective for TGE [12,13,14]. Nonetheless, PEI cytotoxicity counteracts its efficiency and the outcomes of studies of PEI transfection conditions are often contradictory [15,16].

The yield of TGE can be increased by optimizing expression media [17,18] or by preparation of new CHO host lines, expressing various transcription factors, protein chaperones or proteins of unfolded protein response pathways [6,19]. The yield of antibody production following PEI transfection ranged from 0.35 g/L to 2 g/L within 7 days [6,20,21]; however, there are studies that report a much lower yield [22]. To summarize, currently used methods often require cumbersome engineering of host cell line and differ in claimed optimal conditions for gene delivery and cell cultivation.

In this work, we are proposing a fast, flexible, and straightforward procedure for recombinant protein production in existing CHO cell lines. We assembled an expression vector for facile cloning and secretion of expressed proteins and proposed general conditions for PEI-assisted TGE. With this system, we are routinely producing antibodies and human lactoferrin with a yield of up to 100 mg/L within 9 days of cultivation. We optimized the use of recyclable labware, which further reduced the cost of production. Prepared proteins are of high quality, sufficient for biophysical and structural studies, which is demonstrated by diffraction data from crystals of antibody complexes.

## 2. Materials and Methods

### 2.1. CHO Cells Cultivation

ExpiCHO-S cells were cultivated in the ExpiCHO expression medium (ThermoFischer Scientific, Grand Island, NY, USA). After thawing, the cells were handled according to the manufacturer’s instructions and used for transfection after two or three passages. The cells were propagated in the presence of antibiotics (0.06 mg/mL penicillin and 0.1 mg/mL streptomycin, Sigma-Aldrich, St. Louis, MO, USA) at 37 °C and 8% CO_2_ in a humidified atmosphere with constant shaking at 130 rpm on a 19 mm orbital shaker (Advanced Dura-Shaker, VWR Avantor, PA, USA). Non-baffled polycarbonate Erlenmeyer flasks with vent cap (Corning, Corning, NY, USA) or SIMAX glass Erlenmeyer flasks (Kavalierglass, Praha, Czech Republic) with DURAN vented screw cap (DWK Life Sciences, Wertheim, Germany) were used for cell cultivation. The proportion of live and dead cells was determined using the trypan blue dye exclusion method; the cells were counted manually using a hemacytometer. CHO cell line requires handling under at least Biosafety Level 1 containment.

### 2.2. Preparation of Expression Vectors

Universal eukaryotic expression vector pCMV-3UTR was constructed on the base of pCMV-Script (Agilent Technologies, Santa Clara, CA, USA). pCMV-Script was digested with NheI and PvuI restriction endonucleases (New England Biolabs, Ipswich, MA, USA); the Kozak sequence, initiation codon, leader signal peptide (M)EAPAQLLFLLLLWLPDTTG of the human immunoglobulin variable light gene family IGKV3 (Uniprot entry P04433 and homologous light chains), multiple cloning site and 3′ untranslated region (UTR) of the translation elongation factor 1 alpha 1 (EF1 3′ UTR; EEF1A1, NCBI Ref. Seq. NM_001402.6 nucleotides 1452–1655) were inserted (Axon Neuroscience R&D Services, Bratislava, Slovakia), replacing the nucleotides 603–807 of parental pCMV-Script vector.

For the expression of mouse monoclonal antibodies and their Fabs, total RNA was isolated from hybridoma producing DC25, DC11 and MN423 antibodies [23] using the RNeasy Mini Kit (Qiagen, Hilden, Germany). cDNA was synthesized using the Verso cDNA synthesis kit (ThermoFisher Scientific) and antibody variable regions were sequenced following amplification with a library of primers to immunoglobulin leader sequences. Consequently, antibody light and heavy chains were amplified with specific primers using Phusion^®^ Hot Start Flex DNA Polymerase (New England Biolabs), digested with AgeI and XhoI restriction endonucleases (New England Biolabs) and cloned into pCMV-3UTR. DNA of matured human lactoferrin (LF, Uniprot entry P02788 variant 069298), optimized for expression in *Cricetulus griseus* was synthesized via BioCat, Heidelberg, Germany and cloned into pCMV-3UTR as described above for antibody chains. In some experiments, vector pCMV-MCS lacking EF1 3′ UTR was used. All expression plasmids were verified using restriction mapping and/or sequencing. Plasmid pCMV-3UTR is available to academic institutions through Addgene curated collection under the ID 202550.

### 2.3. CHO Cells Transfections

Proteins were produced in 25–50 mL cultures in polycarbonate or glass cultivation flasks. Twenty-four hours before transfection, the cells were transferred to fresh media with a density of 2.3–2.5 × 10^6^ cells/mL. To prepare DNA-PEI polyplexes, DNA stock solution in water was diluted in pre-warmed media in a 15 mL tube and an appropriate volume of PEI stock solution was added. After 7–10 min incubation at room temperature, the DNA-PEI mixture was added to the cell suspension. The final concentration of DNA (total) was 4 μg/mL; the final concentration of PEI was 8 μg/mL. DNA of heavy and light chain of antibodies was co-transfected in an 1:1 ratio. Following transfection, the cells were cultivated at 37 °C and 8% CO_2_ in a humidified atmosphere with constant shaking at 130 rpm on a 19 mm orbital shaker for 24 h. Thereafter, the cultivation conditions were adjusted to 32 °C and 5% CO_2_; 0.9 mM Sodium Butyrate (Sigma-Aldrich) and antibiotics were added. Further, 24 and 120 h after transfection the cultures were supplemented with 4 mL of feeding solution per 25 mL of media. A total of 100 mL of feeding solution was prepared by mixing 70 mL of CHO CD efficient feed A with 14 mL of 170 mg/mL fresh solution of Difco TC Yeastolate ultra-filtered in water, 3.5 mL of 200 mM GlutaMAX (all ThermoFischer Scientific) and 12.5 mL of 45% D (+)-Glucose (PanReac AppliChem, Darmstadt, Germany). The culture medium with secreted recombinant proteins was harvested nine days after transfection.

In some experiments, ExpiCHO-S cells were transfected via electroporation [11]. Twenty-four hours before electroporation the cells were diluted to 2.5 × 10^6^ cells/mL. Immediately prior to electroporation, the cells were resuspended in a HyClone electroporation buffer (MaxCyte, Rockville, MD, USA) to density 2 × 10^8^ cells/mL. DNA was transferred to an empty 1.5 mL microcentrifuge tube (1 μg DNA per 1 × 10^6^ cells); the cell suspension was added, mixed carefully, transferred to OC-400 Processing Assembly and loaded into STX electroporator (MaxCyte). Electroporation was performed using the “CHO” protocol. Cells were transferred immediately to the cultivation flask and recovered at 37 °C and 8% CO_2_ in incubator with a humidified atmosphere for 30 min without shaking. After this period, cells were resuspended in pre-warmed ExpiCHO Expression Medium to final density 5 × 10^6^ cells/mL and antibiotics were added. The cells were cultivated at 37 °C and 8% CO_2_ in a humidified atmosphere with constant shaking at 130 rpm on a 19 mm orbital shaker. Post-transfection cultivation was performed exactly like for PEI transfection.

### 2.4. Purification of Recombinant Proteins

Full-length DC25 (IgG1) and MN423 (IgG2b) antibodies were purified on a 5 mL HiTrap™ Protein A column, DC25 and DC11 Fabs (IgG1) were purified on a 5 mL HiTrap™ protein G column and MN423 Fab (IgG2b) was purified on a 1 mL Protein L column (all chromatography media were from GE Healthcare, Chicago, IL, USA). The final polishing of recombinant Fabs was performed on a HiLoad Superdex 75 16/60 column (GE Healthcare) in 10 mM Tris pH 7.2 and 50 mM NaCl. Fractions were checked on 12% SDS PAGE, pooled and concentrated through ultrafiltration. Purification of recombinant lactoferrin (rhLf) was performed using cation exchange chromatography. The medium of rhLf expressing CHO cells was adjusted to pH 7 by 100 mM NaOH and pre-cleared through centrifugation at 21,000× *g* for 10 min at 4 °C. The supernatant was filtered through a 0.22 mm Acrodisc Syringe Filter and loaded on a HiTrap SP HP 5 mL column (GE Healthcare) primed using a mobile phase of 20 mM sodium phosphate pH 7. Protein elution was accomplished with a gradient of 1.5 M NaCl in the mobile phase buffer, and rhLf was eluted as a symmetric peak fraction at about 0.7 M NaCl. The concentration of pure proteins was determined from the absorbance at 280 nm.

### 2.5. Dynamic Light Scattering

All pipette tips and tubes were cleaned from dust particles via filtered argon stream. Purified antibody Fabs were diluted to a concentration of 1–5 mg/mL. Briefly, 10 μL of Fab solution in a 0.5 mL microcentrifuge tube was centrifuged at 5000× *g* for 5 min at room temperature to sediment dust particles and potential protein aggregates. Briefly, 4 μL of the sample was transferred to a disposable MicroCuvette (Wyatt Technology, Santa Barbara, CA, USA) and measured in the Dynapro Nanostar (Wyatt Technology). Measurements were performed at 25 °C with 7–10 s acquisition time averaged 5–10 times. Data from at least five individual measurements of dynamic light scattering (DLS) per sample were evaluated using the Dynamics software v. 7.10.0.21. To cull the acquisitions influenced by dust or irregular particles, an automatic filtering of autocorrelation functions was applied with an individual limit for baseline threshold and maximal allowed sum-of-squares (SOS) error for cumulants fit. After filtering, at least 65% of the original data remained for analysis. To determine the hydrodynamic radius, polydispersity and molecular weight, cumulants analysis using the Dynals algorithm implemented in Dynamics software was used. Molecular weight was extrapolated using a model for globular proteins. Parameters are expressed as the means and their standard deviations.

### 2.6. Crystallization of the Recombinant Antibodies and Their Complexes with Tau 297–391

DC11 Fab crystals have been grown in sitting drop using the condition B3 of PACT Premier screen (Molecular Dimensions, Rotherham, UK) with a final protein concentration of 8 mg/mL. Recombinant protein tau 297–391 (dGAE) corresponding to the tau protein found in the core of Alzheimer’s disease pathologic filaments [24,25,26,27] was prepared as described previously [28,29]. Antibody-dGAE complexes were prepared just before crystallization experiments. dGAE was incubated for 30 min at room temperature with 1 mM DTT to obtain dGAE monomers. Binary complex of MN423 Fab with dGAE (complex I) and ternary complex of DC11 Fab, DC25 Fab and dGAE (complex II) were prepared by mixing proteins in an equimolar ratio; the final concentration of DTT was less than 100 μM. The mixture was incubated for 15 min at room temperature following complex isolation via size exclusion chromatography on a Superdex 200 10/30 column (Cytiva, Uppsala, Sweden) equilibrated in 10 mM Tris-HCl pH 7.2, 50 mM NaCl. Eluted complexes were concentrated using Amicon Ultra concentrators with 10 kDa cut off (Merck, Darmstadt, Germany) and stored at −20 °C. The concentration of proteins was deduced from the chromatogram peak area. For screening of the crystallization conditions in siting drop, the Proplex, PACT Premier and LMB screens were used (Molecular Dimensions). Screening was performed on 96-well Swisssci UVXPO MRC Plate (Molecular Dimensions) through manual dispensing [30] or with NT8 Drop Setter (Formulatrix, Bedford, MA, USA). Crystallization plates were inspected with a RockImager 54 (Formulatrix) at 22 °C and crystallization was assessed using visual evaluation. Complex I was crystallized from hanging drop in 24-well EasyXtal plates (Nextal Biotechnologies, Holland, OH, USA) using condition B5, PACT Premier screen with final protein concentration 4 mg/mL. Complex II was crystallized in sitting drop using condition F1, LMB screen with final protein concentration 5 mg/mL.

### 2.7. Diffraction Experiment and Data Processing

Crystals were fished out with nylon loops, cryoprotected in Paraton N (Hampton Research, Aliso Viejo, CA, USA) and flash-cooled in liquid nitrogen. Diffraction data were collected at beamline P13 (EMBL DESY, Hamburg, Germany) and PXI (SLS PSI, Villigen, Switzerland) for DC11 Fab, complex II, and complex I, respectively. Individual diffractions were indexed, integrated and scaled in XDS package [31] Laue group was determined via POINTLESS and data were merged in AIMLESS [32].

## 3. Results

### 3.1. Optimization of the Platform for Transient Gene Expression

First, we aimed to construct an effective expression plasmid. Our criteria were (i) flexible cloning of coding DNA, (ii) export of protein product extracellularly, (iii) optimal size facilitating gene delivery, and (iv) suitable selection marker. We took account of the elements enhancing the amount and stability of mRNA. Novel expression vector pCMV-3UTR (Figure 1A) has been constructed using the backbone of commercially available pCMV-Script vector. It features a human cytomegalovirus immediate early promotor resulting in one of the highest levels of expression of target protein mRNA [33], a Tn5 aminoglycoside phosphotransferase ORF coding for a kanamycin/neomycin resistance marker, which enables selection in bacteria through kanamycin and creation of stably expressing mammalian cell lines with G418 selection, and a pUC origin of replication ensuring a high plasmid copy number in bacteria. The expression cassette of pCMV-3UTR (Figure 1B, nucleotides 603–892) is composed of Kozak sequence, a signal peptide driving the secretion of produced proteins into a cultivation media, and multiple cloning site (MCS) containing AgeI, SpeI, EcoRI and XhoI recognition sequences (for isoschizomers and enzymes with alternative sticky ends, see Appendix A). The expression cassette ends with 3′ UTR of human EEF1A1.

The last two codons of the signal peptide (amino acids Thr-Gly) were modified in a way to create an AgeI restriction site ACCGGT. This offers large flexibility for seamless insertion of the protein coding sequence into the correct reading frame. Moreover, AgeI 5′ overhang sequence is complementary to seven compatible sticky ends created by 20 alternative restriction endonucleases (Appendix A), which allows for cloning through this site even when the AgeI would cut inside the cloned sequence.

Insertion of 3′ UTR influences the stability of transcribed RNA and may increase the protein yield [34,35]. To see if the 3′ untranslated RNA may increase protein production from pCMV-3UTR plasmid, we used Fab of monoclonal antibody DC25, which recognizes all isoforms of neuronal protein tau, binding to a linear epitope 347KDRVQSK353 in the repeat region of tau protein [23]. Using electroporation for gene delivery, we compared DC25 Fab expression yield using pCMV-3UTR and pCMV-MCS plasmid lacking the EEF1A1 3′ UTR (Figure 1C). Average DC25 Fab expression from pCMV-3UTR (125.9 ± 19.2 mg/L) was significantly higher than production from pCMV-MCS (36.4 ± 10.5 mg/L), which underlined the importance of 3′ UTR for protein production.

One of the factors influencing the cost of the protein production cycle, its effectiveness and impact on the environment is the cultivation container. The prevalently used cultivations flasks are disposable ones made from polycarbonate. To further optimize the production process, we investigated the cultivation of CHO cells in recyclable glass cultivation flasks, comparing them to polycarbonate flasks. Cells cultivated in both types of material in parallel were passaged every 2–3 days during the 19 days of the experiment and regularly checked for doubling time and viability (Figure 1D). We found no significant difference in relation with the cell culture health. For the cells cultivated in glass flasks, the doubling time reached 17.8 ± 0.8 h and viability 97.8 ± 0.5%, while for cultivation in the polycarbonate flasks, the doubling time was 17.5 ± 1.0 h and viability 97.6 ± 0.6%. We can conclude that the material of cultivation flasks does not have an impact on cell proliferation and viability.

### 3.2. Production of Recombinant Proteins

We have cloned several antibodies and human lactoferrin into pCMV-3UTR and monitored their expression in CHO cell line (Figure 2). In addition to the Fab and full-length IgG1 mouse antibody DC25 described in Section 3.1, we also expressed the Fab of mouse antibody DC11 specific to a conformation of tau protein found in the Alzheimer’s disease pathology [36,37], as well as the Fab and full-length IgG2b mouse antibody MN423, recognizing a conformational epitope in the core of Alzheimer’s disease tau filaments [38]. We also expressed human iron-containing glycoprotein lactoferrin, which is an important player in innate immune response [39]. Analysis of expression at several post-transfection days on SDS-PAGE showed a steady state increase in expressed proteins (Figure 2A); however, not all proteins expressed equally well. The most difficult to express was the full-length antibody MN423 of IgG2b isotype, also providing the lowest yield after purification on affinity chromatography. It has been previously shown that the specific nucleotide sequences may heavily influence the yield of TGE of antibodies [6]. The same may hold for expression of different antibody isotypes; it was shown that some antibody classes represent obstacles in CHO cells production as they accumulate in the lumen of endoplasmic reticulum, where one of the chain aggregates or the disulfide bonds do not form [40,41]. We stopped the cultivation on day 9 when the viability of cells decreased rapidly. The yield of PEI-assisted TGE from pCMV-3UTR plasmid in CHO cells ranged from 8 to 100 mg/L (Figure 2B). Purity of Fabs and lactoferrin after two-step and one-step chromatography purification, respectively, was very high (Figure 2A).

To benchmark the process of TGE by PEI gene delivery, we produced Fabs of DC25, DC11, MN423 and full-length MN423 and DC25 after gene delivery via electroporation as well (Figure 2B). The methods are comparable: PEI gene delivery provided even higher yield for DC11 Fab and full-length DC25; MN423 Fab and DC25 Fab expressed better after electroporation. Full-length MN423 expressed only slightly better via electroporation.

### 3.3. Crystallography of Alzheimer’s Disease Tau Protein Using Recombinant Antibodies

Dynamic light scattering measurement revealed expected molecular weight and very low polydispersity, meaning that the antibodies are highly homogenous (Table 1). This encouraged us to use them for crystallization experiments to solve their structure and the structure of their complexes with tau protein antigen.

Antibodies DC25 Fab and MN423 Fab have been crystallized previously [42,43]; therefore, we focused on the crystallization of DC11 Fab and complexes with tau protein. Particularly, as DC11 and MN423 recognize conformational epitopes on tau protein folded in disease-related filaments in Alzheimer’s disease [36,38], we aimed to co-crystallize these antibodies with recombinant fragment of intrinsically disordered tau protein dGAE, found in the core of Alzheimer’s filaments [24]. It has been suggested that tau protein in complexes with conformation-specific antibodies may acquire a disease-specific conformation, which can be solved from X-ray diffraction data [43,44]. To this end, we prepared two complexes: MN423 Fab + dGAE (Complex I) and DC11 Fab + DC25 Fab + dGAE (Complex II). Antibody DC25 Fab, which does not compete with DC11 for binding to tau, was included in ternary Complex II as a crystallization chaperon to facilitate crystal growth by creating more crystal contacts. The stability of both complexes was high enough to withstand purification via size exclusion chromatography (Figure 3A).

DC11 Fab formed monoclinic crystal with a size of 100 × 20 × 10 μm^3^ in 0.1 M MIB, pH 6, and 25% (*w*/*v*) PEG 1500 (Figure 3B). After initial screening of the crystallization conditions, monoclinic crystals of Complex I with a size of 100 × 40 × 10 μm^3^ grew in 0.1 M MIB, 8 pH, and 25% (*w*/*v*) PEG 1500 (Figure 3C). Orthorhombic crystals of the Complex II with a size of 110 × 20 × 10 μm^3^ appeared in the condition of 0.2 M ammonium sulfate, 0.1 M sodium acetate, pH 5.6, and 25% (*w*/*v*) PEG 4000 (Figure 3D). Details of the crystallization conditions are summarized in Appendix A. Diffraction data of crystals were collected at the synchrotron source of X-ray radiation and complete data sets were collected at 1.3–2.7 Å resolution (Appendix A).

## 4. Discussion

Despite extensive efforts undertaken in the development of methods for recombinant protein production via TGE in mammalian cells, it is difficult to find a universal protocol for this task suitable for small research projects. Due to the large significance of recombinant proteins, namely antibodies for pharmacotherapy development in the pharma industry, majority of the methods are being developed using industrial resources and technologies. In this paper, we developed a novel procedure focused on the flexibility of use with a minimal cost and demonstrated the production of various antibody Fabs and human lactoferrin in milligram quantities using ExpiCHO-S cell line. The method is cost-effective, as it needs modest initial investments and exhibits a low cost per transfection (Appendix A). The minimal starting investment adds up to only 60% of the minimal cost for a transfection kit using polycationic lipid transfection reagent, and the cost of 25 mL of transfected cell culture is more than 14 times lower, totalling 2.5 EUR per one transfection with our PEI-based procedure. The costs and impact on the environment are further reduced by using recyclable glassware, which can be reused almost unlimitedly. Our system can be easily adopted even by small research teams with a limited budget.

Novel eukaryotic expression vector pCMV-3UTR allows protein transcription from CMV promotor, which is one of the strongest among viral or endogenous promotors tested in CHO cells [33,45]. Various antibodies and other proteins can be seamlessly cloned using AgeI site, introduced at the end of the immunoglobulin signal peptide. It was shown that the protein production in CHO cells can be enhanced by optimizing signal peptide sequences [46,47]; however, we opted for signal peptide allowing easy cloning at the expense of a potentially lower yield. The selection of a tailored signal peptide in future variants of vector offers space for further development. The relatively small size of the new vector allows for lower concentration of PEI at transfection (at PEI/DNA ratio equal 2:1), which is beneficial due to the PEI toxicity [48].

Our results showed a high homogeneity of recombinant Fabs, which is an important prerequisite for successful protein crystallization. Indeed, we were able to obtain crystals diffracting to high resolution, assuring sufficient structural details in solved structure. Fabs prepared using traditional papain digestion of full-length antibodies have an intrinsic variability at the digested C-terminus of the heavy chain, increasing the heterogeneity of final protein product, which can potentially jeopardize the crystallization process. Moreover, it was hypothesized that papain used in antibody digestion may co-purify with Fabs and remain proteolytically active even in the crystallization drop [49]. This is not the case with the recombinant Fabs. Recombinant expression of lactoferrin showed the potential of the platform to produce glycosylated metalloproteins.

Developed TGE platform was able to produce milligrams of recombinant Fabs and lactoferrin in a small-scale culture within nine days, using cost-effective transfection agent and reusable laboratory material. The yield and cost can be further optimized by titrating the PEI/DNA ratio, altering cell density or media composition [17,50].

## Figures and Tables

**Figure 1 antibodies-12-00051-f001:**
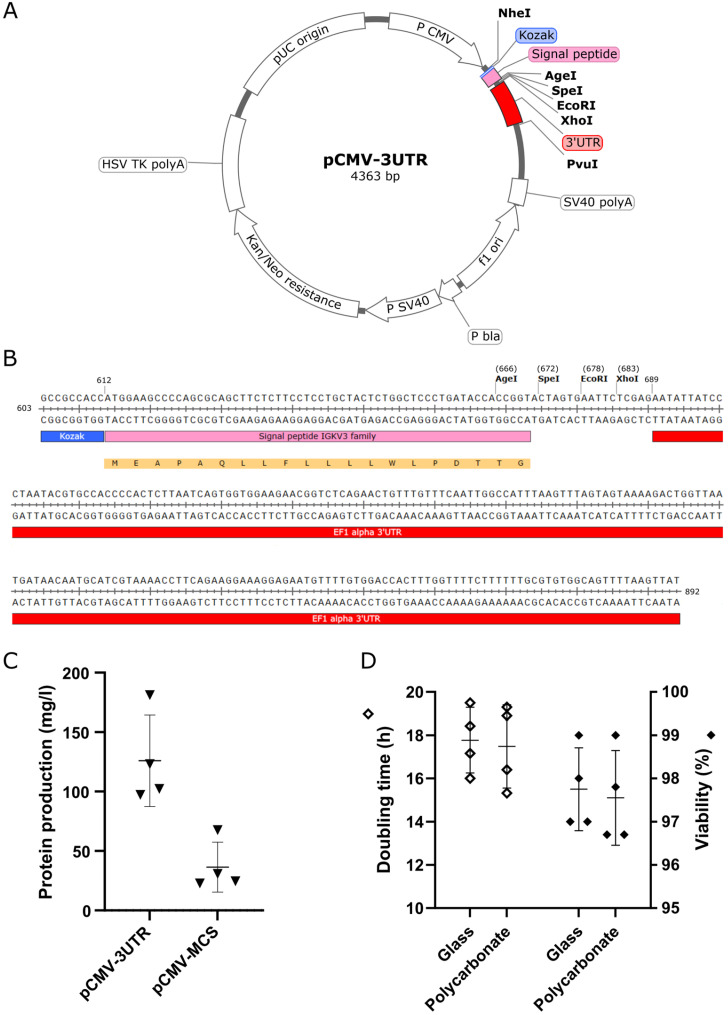
Development of expression platform. (**A**) Map of expression vector pCMV-3UTR, highlighting its individual components. The expression cassette contains Kozak sequence in blue, signal peptide in pink and 3′ UTR in red. NheI and PvuI sites were used for the cloning of the cassette into the parental pCMV-Script vector. Positions of individual cloning sites in the cassette are indicated as well. Designed in SnapGene Viewer v. 7.0.1. (**B**) Expression cassette of pCMV-3UTR. Colour coding as in panel A. Translated sequence of signal peptide is shadowed yellow. (**C**) The level of DC25 Fab expression in Chinese Hamster Ovary (CHO) cells from the plasmid with (pCMV-3UTR) or without (pCMV-MCS) the 3′ UTR. (**D**) Comparison of doubling times and viability of CHO cells cultivated in flasks made of glass or polycarbonate. In (**C**,**D**) the mean and standard deviation are indicated.

**Figure 2 antibodies-12-00051-f002:**
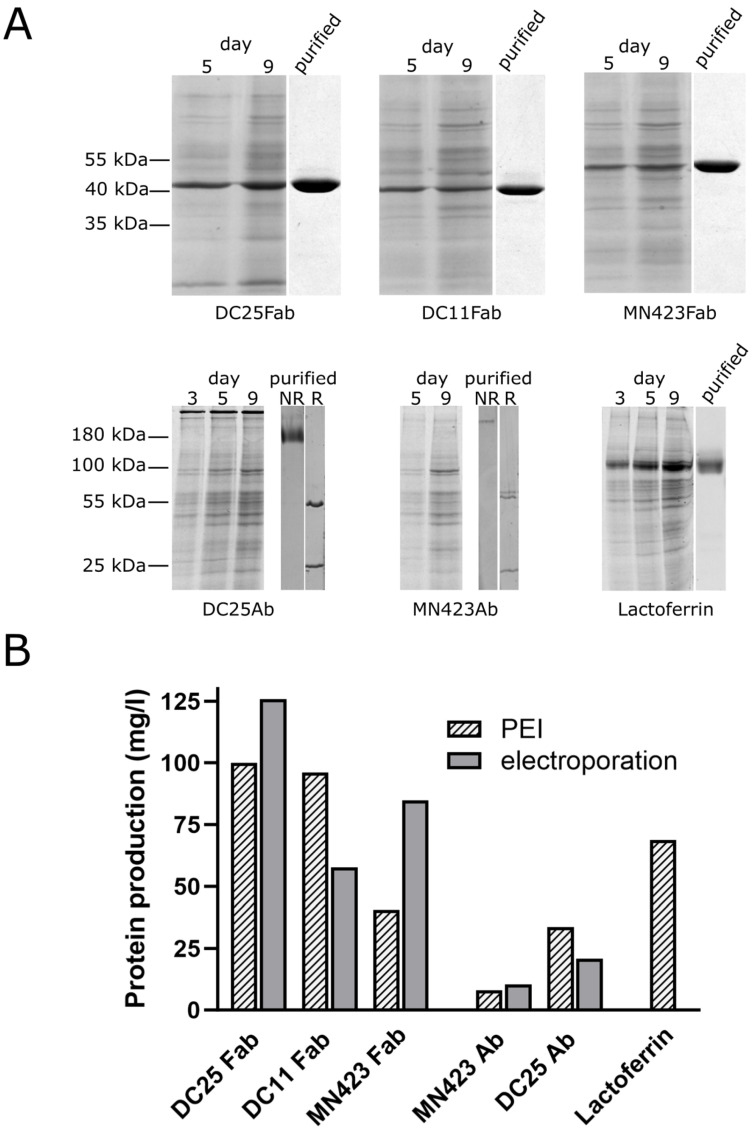
Validation of transient gene expression from pCMV-3UTR plasmid in CHO cells after polyethylenimine (PEI) assisted gene delivery. (**A**) Coomassie-stained 12% SDS polyacrylamide gels after electrophoresis of cultivation media samples and/or purified proteins. The post-transfection day is indicated above the gels. All samples were analysed in non-reducing conditions, except the purified full-length DC25 and MN423, where the reduced samples were included as well, showing the positions of light chains (~25 kDa) and variably glycosylated heavy chains (~55 kDa). NR—non-reduced; R—reduced. (**B**) The yield in protein production from pCMV-3UTR plasmid in CHO cells after transfection through PEI and electroporation. Lactoferrin was expressed only through PEI transfection.

**Figure 3 antibodies-12-00051-f003:**
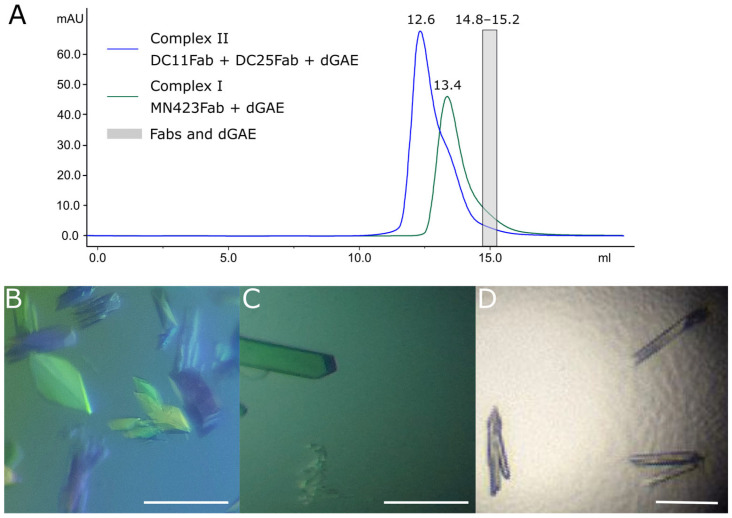
Crystallization of DC11 Fab and complexes of antibodies with tau dGAE. (**A**) Isolation of dGAE-Fab complexes via size exclusion chromatography. Retention volumes are indicated above the peaks. Retention volume of monomeric Fabs and dGAE are in the range of the grey box. (**B**) Crystals of the DC11Fab; (**C**) crystals of the binary Complex I of MN423 Fab and dGAE; (**D**) crystals of the ternary Complex II of DC25 Fab, DC11 Fab and dGAE. Scale bar in (**B**–**D**) represents 100 μm.

**Table 1 antibodies-12-00051-t001:** Dynamic light scattering characterization of purified recombinant Fabs.

Antibody	Radius [nm]	PD [%]	MW [kDa]
DC11Fab	2.952 ± 0.012	2.3 ± 0.7	42
MN423Fab	3.051 ± 0.009	4.7 ± 1.7	46
DC25Fab	2.961 ± 0.012	1.9 ± 0.3	43

## Data Availability

Data is contained within the article or Appendix A.

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
