# Peer review of "Cost-Effective Protein Production in CHO Cells Following Polyethylenimine-Mediated Gene Delivery Showcased by the Production and Crystallization of Antibody Fabs"

_2073-4468, 2023, doi:10.3390/antib12030051_

Round 1
Reviewer 1 Report
The research study follows a rational scientific design overall and interpreted the results are very well written and well-presented and discussed, though there is a need to adjust some of the claims made throughout the text for the methodology in terms of being time- and cost-effect or how universal and applicable it may be for others to repeat it.
Here are my specific comments:
1. The title needs a bit of modifications as the time spent for the expression (average 9 days reported here) is longer than the general CHO expression (about 5 days). The cost also may not be that much different at least for small scale expression. So, I am not sure if this methodology which worked great for some proteins and antibodies to certain extent is really time and cost -effective.
2. Line 75: the claim of 100 mg/ ml for the yield contradicts the statement in line 283-284 (8-100 mg/l).
3. Line 11-112: is there any reason why in some experiments vector pCMV-MCS lacking EFI 3’UTR was used?”
4. Figure 2B: please give an explanation on why only PEI used for the transfection of Lactoferrin?
5. Line 334: I think there is an error in citing “Table SI: which should be “Table SII)
6. Line 344: please elaborate the main novel elements of your systems that could be distinguished from what has already been published. Most of the sequence elements are not novel but their assembly could be and this needs to be explained and highlighted well as it is main objective of this manuscript.
7. Finally, I did not see any safety statement for the cell line handling and that fact that all these works need to be done at Biosafety level II or higher in particular for handling the CHO cell lines.
Author Response
Please see the attachment. Thank you for your effort.

Reviewer 2 Report
In this manuscript by Klaudia Meskova, et al., entitled " Time- and cost-effective antibody production in CHO cells following polyethylenimine-mediated gene delivery" reported that for transient expression of mainly Fab in CHO cells, 25-50 mm-scale cultures with PEI were examined and milligram-scale yields were obtained. In fact, even protein complex crystals were obtained with the obtained Fab, indicating that this method is useful for protein expression. However, the title does not seem to match the content. My remarks on this manuscript are presented below.
1. Although commercially available ExpiCHO-S cells and culture media (ExpiCHO Expression System) are used, there is no comparison of cost and time when using original ExpiFectamine CHO Transfection reagents.
2. I thought the most important aspect of this paper was the successful construction of a novel vector pCMV-3UTR, but I could not understand why the authors included the EF1 3' UTR sequence. Please describe any prior literature or evidence as to why the sequence to be inserted in the 3' UTR is EF1 and not another gene.
3. On line 384, it says “Our results demonstrate the high homogeneity of the recombinant fabs, an important prerequisite for successful crystallization." but since it is purified, doesn't this new method of expression matter?
4. While examples of full-length antibodies and lactoferrin are listed, there is no consideration of downstream applications such as Fab. It would be easier to understand if the title focused on Fab and omitted examples of antibodies and lactoferrin.
5. Line 257 says “Cultures were maintained in parallel for 19 days”, is this correct?
6. Is the reason for 9 days culture in TGE because protein production does not rise any further? Please provide any evidence, such as data on survival rates and protein production over time.
Author Response

(The authors gave the same response as above.)

Round 2
Reviewer 2 Report
Thank you for sending me the revise manuscript.
I think the manuscript adequately answers the points raised. 
I recommend that it be accepted for publication.
Author Response
We are grateful to the Reviewer 2, we are sure that s/he comments and corrections improved the manuscript.
Yours sincerely
Rostislav Skrabana